∂ | **Open Peer Review** | Mycobacteriology | Research Article

# Ultra-high field strength electroporation enables efficient DNA transformation and genome editing in nontuberculous mycobacteria

**Daoyan Tang,[1] Minggui Wang,[2] Dan Wang,[2] Danni Yang,[2] Yi Cai,[3] Tao Luo,[4] Jianqing He,[1,5] Qinglan Wang[2,5]**

**ABSTRACT**    Efficient DNA delivery is essential for genetic manipulation of mycobacteria and for dissecting their physiology, pathogenesis, and drug resistance. Although electroporation enables transformation efficiencies exceeding $10^5$ CFU per µg DNA in *Mycobacterium smegmatis* and *Mycobacterium tuberculosis*, it remains highly inefficient in many nontuberculous mycobacteria (NTM), including *Mycobacterium abscessus*. Here, we discovered that NTM such as *M. abscessus* exhibit exceptional tolerance to ultra-high electric field strengths and that hypertonic preconditioning partially protects cells from electroporation-induced damage. Using ultra-high electric field strength (3 kV/mm) electroporation, we achieved dramatic improvements in plasmid transformation efficiency—up to 106-fold in *M. abscessus*, 83-fold in *Mycobacterium marinum*, and 37-fold in *Mycobacterium kansasii*—compared to standard conditions (1.25 kV/mm). Transformation efficiency was further influenced by the choice of selectable marker. Ultra-high field strength electroporation also markedly enhanced allelic exchange in *M. abscessus* expressing Che9c RecET recombinases, increasing the recovery of gene deletion mutants by over 1,000-fold relative to conventional electroporation. In parallel, oligonucleotide-mediated recombineering for targeted point mutations produced nearly 10,000-fold more mutants under ultra-high field conditions. Together, these findings establish ultra-high field electroporation as a robust, broadly applicable platform for genetic engineering of NTMs. This method substantially enhances transformation efficiency and enables construction of advanced genetic tools—including expression libraries and CRISPRi knockdown libraries—in species that have historically resisted genetic manipulation.

**IMPORTANCE** Infections caused by nontuberculous mycobacteria (NTM), including *Mycobacterium abscessus*, are increasing globally, yet genetic manipulation of these pathogens remains technically challenging due to inefficient DNA delivery and low gene editing success. The ultra-high electric field strength electroporation strategy described here overcomes these barriers, enabling dramatic improvements in both transformation and genome editing efficiency. This advance paves the way for high-throughput functional genomics in NTMs, including the construction of genome-wide knockout, CRISPRi knockdown, and expression libraries. Broad adoption of this approach will accelerate discovery of genetic determinants of virulence and drug resistance, facilitating the development of antimicrobials and vaccines.

**KEYWORDS**   Ultra-high field strength electroporation, DNA transformation, genome editing, nontuberculous mycobacteria, *Mycobacterium abscessus*

Address correspondence to Qinglan Wang, wangqinglan@scu.edu.cn, Jianqing He, jianqing_he@scu.edu.cn, or Jianqing He, jianqing_he@scu.edu.cn.

Daoyan Tang and Minggui Wang contributed equally to this article. Author order was determined by mutual agreement among the authors.

The authors declare no conflict of interest.

See the funding table on p. 11.

The genus *Mycobacterium* comprises nearly 200 species, among which only a few —*Mycobacterium tuberculosis (M. tuberculosis)*, *M. leprae*, and *M. lepromatosis*—are

traditionally associated with tuberculosis or leprosy. The remainder, collectively termed nontuberculous mycobacteria (NTMs), are ubiquitous in natural environments, and several species are opportunistic pathogens capable of causing pulmonary, soft tissue, and disseminated infections (1). Among NTMs, *Mycobacterium abscessus (M. abscessus)* has emerged as a clinically significant pathogen. Once primarily associated with cystic fibrosis, *M. abscessus* now may cause both pulmonary and extrapulmonary infections in patients without underlying disease (2, 3). The species is notoriously recalcitrant to therapy, with treatment success rates of only 20–50% despite prolonged multidrug regimens (4), leading to its designation as an "incurable nightmare" (5, 6).

Genetic manipulation of mycobacteria is central to advancing our understanding of their physiology, pathogenic mechanisms, and drug resistance. However, the notoriously low permeability of the mycobacterial cell wall severely hampers DNA uptake via electroporation—the most widely used transformation method (7, 8). While relatively high transformation efficiencies ($10^5$–$10^6$ CFU/µg DNA) have been reported in *M. smegmatis* and *M. tuberculosis*, transformation in NTMs remains inefficient and variable across species. For example, *M. abscessus* typically yields fewer than $5 \times 10^4$ CFU/µg DNA when transformed with kanamycin-resistant plasmids, and *M. kansasii* is even less amenable to transformation, often producing fewer than 10 CFU/µg DNA under comparable conditions (9–11).

Homologous recombination using the mycobacteriophage Che9c RecET system enables more efficient genome editing than endogenous pathways (12, 13). RecE functions as a 5'→3' exonuclease that resects double-stranded DNA, producing 3' overhangs that are subsequently bound by RecT, thereby facilitating strand annealing and homologous exchange (13). While RecET recombineering is effective in *M. smegmatis* and *M. tuberculosis*, it has performed poorly in *M. abscessus*, where recombineering yields few antibiotic-resistant colonies, and only approximately 7% of these harbor the desired mutation (14). Similarly, oligo-mediated recombineering efficiencies in *M. abscessus* are orders of magnitude lower than those in *M. smegmatis*, often falling below 1 in $10^6$ (12, 15, 16), likely reflecting poor DNA uptake.

These limitations severely restrict the use of reverse genetics, CRISPR interference libraries, and expression screens in NTMs. To overcome this barrier, we sought to optimize electroporation conditions for DNA delivery in both *M. abscessus* (ATCC 19977) and *M. marinum* (M strain). Using a range of resistance markers, we systematically tested key electroporation parameters and identified ultra-high electric field strength as a critical factor in enhancing transformation efficiency. This strategy significantly improves DNA uptake and gene editing capacity in NTMs, laying the groundwork for more effective genetic manipulation in these clinically relevant pathogens.

## MATERIALS AND METHODS

### Strains, media, and reagents

The *M. abscessus* ATCC 19977 wild-type strain, *M. smegmatis* mc²155, and *M. marinum* M strain were generously provided by Prof. Tao Luo (Sichuan University). A clinical isolate of *M. kansasii* was obtained from West China Hospital, Sichuan University.

*M. marinum* was cultured in Middlebrook 7H9 broth supplemented with 0.2% (vol/vol) glycerol, 0.01% (vol/vol) tyloxapol, and 10% (vol/vol) OADC enrichment (OADC formulation per 100 mL: 5 g bovine serum albumin, 2 g glucose, 0.81 g NaCl, and 60 µg oleic acid, adjusted to 100 mL with deionized water and sterile-filtered). All other mycobacteria were cultured in Middlebrook 7H9 broth supplemented with 0.2% (vol/vol) glycerol, 0.01% (vol/vol) Tween 80, and 10% (vol/vol) OADC. For solid culture, strains were grown on Middlebrook 7H10 agar supplemented with 0.5% (vol/vol) glycerol and 10% (vol/vol) OADC. Antibiotics were used at the following final concentrations: for *M. abscessus* and *M. kansasii*, kanamycin at 100 µg/mL and zeocin at 25 µg/mL; for *M. marinum* and *M. smegmatis*, kanamycin at 25 µg/mL and hygromycin at 50 µg/mL.

Kanamycin, hygromycin, ofloxacin, lithium acetate (LiAc), dithiothreitol (DTT), sorbitol, mannitol, and dimethyl sulfoxide (DMSO) were purchased from Aladdin (Shanghai, China). Zeocin was obtained from Thermo Fisher Scientific (California, USA). Bovine serum albumin (BSA), sodium chloride (NaCl), glucose, oleic acid, glycine, glycerol, acetamide, Tris-HCl, and Tween 80 were sourced from Sangon Biotech (Shanghai, China). Tyloxapol was purchased from MedChemExpress (New Jersey, USA). Middlebrook 7H9 and 7H10 media were obtained from Becton, Dickinson and Company (BD; New Jersey, USA).

## Preparation of mycobacterial competent cells

### Conventional competent cell preparation

Mycobacteria were grown from 1 mL of frozen stock inoculated into 100 mL of Middlebrook 7H9 broth supplemented with 0.2% glycerol, 0.01% Tween 80 or tyloxapol (for *M. marinum*), and 10% OADC. Cultures were incubated with agitation until mid-log phase ($OD_{600} = 0.6$–$0.8$), then processed as follows:

For *M. smegmatis*, 100 mL cultures were incubated at 37°C with shaking at 180 rpm. At mid-log phase, cultures were chilled on ice for 1.5 h, pelleted by centrifugation (4°C, 4,000 rpm, 10 min), and washed sequentially with 30 mL, 20 mL, and 10 mL of ice-cold 10% glycerol. Cells were finally resuspended in 1 mL of 10% glycerol.

For *M. abscessus*, 100 mL cultures were incubated at 37°C, 180 rpm, followed by induction with 0.2 M glycine for 3 h. For the "ice group," cultures were placed on ice for 1 h before centrifugation at 4°C. For the "room temperature group," cells were collected directly at room temperature. Subsequent washing steps were identical to those for *M. smegmatis*.

For *M. marinum*, 100 mL cultures were grown at 32°C with shaking at 100 rpm and induced with 0.2 M glycine for 4 h. "Ice" and "room temperature" groups were processed as described for *M. abscessus*.

For *M. kansasii*, 100 mL cultures were incubated at 37°C, 100 rpm, and induced with 0.2 M glycine for 24 h, followed by centrifugation at room temperature. Washing and resuspension steps followed the same procedure as for *M. smegmatis*.

### Special treatments for competent cells

DMSO treatment: Competent cells prepared as above were treated with DMSO at final concentrations of 1%, 2%, or 3%. DMSO was added immediately before electroporation, followed by a 10-minute incubation with plasmid DNA. For the "2% DMSO recovery" condition, 2% DMSO was also included in the recovery medium.

LiAc/DTT treatment: After harvesting from 20 mL of cultures, cells were resuspended in 3 mL of buffer containing 10 mM Tris-HCl (pH 7.5), 10% glycerol, 100 mM lithium acetate, and 10 mM dithiothreitol. Suspensions were incubated at 37°C for 30 min with shaking, then washed and resuspended in 10% glycerol as described above.

42°C heat shock: Cultures were incubated in a 42°C water bath for 1 h at mid-log phase. Subsequent processing followed the conventional protocol.

### Hyperosmotic competent cell preparation

Cultures were grown in hyperosmotic 7H9 medium supplemented with 0.5 M sorbitol to $OD_{600} = 0.6$–$0.8$, followed by induction with 0.2 M glycine. Cells were harvested and washed three times with a buffer containing 10% glycerol, 0.5 M sorbitol, and 0.5 M mannitol. The final pellet was resuspended in 1 mL of the same buffer.

## Electroporation and recovery of mycobacteria

### Conventional electroporation

Electroporation was performed by mixing 1 µg of desalted plasmid DNA with 200 µL of competent cells. The mixture was transferred to a 2 mm gap cuvette and incubated on

ice for 10 min (room temperature for species other than *M. smegmatis*). Electroporation was carried out using a MicroPulser electroporator (Bio-Rad) at 2.5 kV in a 2 mm gap cuvette (equivalent to 1.25 kV/mm field strength). Cells were immediately transferred to 3 mL of 7H9-glycerol-Tween-OADC recovery medium, and the cuvette was rinsed to maximize cell recovery.

Recovery conditions were as follows: *M. smegmatis*: 37°C, 180 rpm, 2 h; *M. abscessus*: 37°C, 180 rpm, 4 h; *M. marinum*: 32°C, 100 rpm, 4 h; *M. kansasii*: 37°C, 100 rpm, 24 h. Post-recovery, cells were plated on 7H10-glycerol-OADC plates containing the appropriate antibiotic. Plates were incubated at: *M. smegmatis* at 37°C for 3 days, *M. abscessus* at 37°C for 4 days, *M. marinum* at 32°C for 7 days, and *M. kansasii* at 37°C for 20 days.

### High-field strength electroporation

For high-voltage electroporation, 1 µg of water-eluted plasmid DNA was added to 100 µL of competent cells in a 1 mm gap cuvette. Care was taken to eliminate air bubbles and ensure the outer cuvette surface was dry to prevent arcing.

Voltage gradient experiments (using hyperosmotically prepared cells) were conducted at 2.1, 2.4, 2.7, and 3.0 kV (field strengths of 2.1–3.0 kV/mm). Post-electroporation, cells were recovered in 7H9-glycerol-Tween-OADC supplemented with 0.5 M sorbitol and 0.38 M mannitol. Recovery conditions followed those described in Section "Conventional electroporation".

For conventionally prepared cells, electroporation was conducted at 3.0 kV (3.0 kV/mm), and cells were recovered in standard 7H9-glycerol-Tween-OADC medium as described above.

### Gene knockout in *M. abscessus*

Two milliliters of *M. abscessus* harboring the pJV53-zeo-mScarlet plasmid were inoculated into 200 mL of 7H9-glycerol-Tween-OADC medium. Cultures were grown to mid-log phase, induced with 0.2 M glycine for 3 h, and then with 0.2% acetamide for an additional 4 h. Cells were harvested and made electrocompetent as described above. For allelic exchange, linear double-stranded DNA substrates (~1 kb homology arms flanking a *kanR* cassette) were digested with restriction enzymes and introduced via electroporation. Cells were recovered at 37°C with shaking for 12 h and then plated on 7H10-glycerol-OADC-kanamycin plates. After 4 d incubation at 37°C, colonies were screened by PCR and Sanger sequencing.

### Site-directed mutagenesis

Site-specific mutagenesis was performed in *M. abscessus* harboring the pKM461-dkan-ZeoR plasmid. Two milliliters of seed culture were inoculated into 200 mL of 7H9-glycerol-Tween-OADC and grown to mid-log phase. Cultures were induced with 0.2 M glycine for 3 h, followed by 500 ng/mL anhydrotetracycline for 4 h. Cells were collected and made competent. Single-stranded DNA oligonucleotides (Oligo 1 or Oligo 2) were electroporated individually into competent cells. After recovery at 37°C for 12 h, cells were plated onto 7H10-glycerol-OADC-kanamycin plates. The editing efficiency of each oligo was assessed by colony counts.

To introduce a resistance mutation (*gyrA*Asp96Asn), a corresponding oligo was electroporated, followed by recovery and plating onto 7H10-glycerol-OADC plates containing 80 µg/mL ofloxacin. Plates were incubated at 37°C for 4 days. Colonies were screened by PCR and sequencing for confirmation.

## RESULTS

### Optimizing electrotransformation in *M. abscessus* and *M. marinum* via cell wall–weakening strategies

Efficient genetic manipulation of NTMs remains challenging due to poor electrotransformation efficiency, which is largely attributed to the low permeability of the mycobacterial cell envelope. To address this, we systematically evaluated several pretreatment strategies in *M. abscessus* and *M. marinum,* including glycine supplementation, membrane-permeabilizing reagents, and heat shock (7, 10, 17–22).

Among all tested conditions, glycine pretreatment proved the most robust. Supplementation with 0.2 M glycine increased transformation efficiency of plasmids pQL037 (kanamycin resistance) and pQL038 (zeocin resistance) in *M. abscessus* by 16-fold and 7-fold, respectively (Fig. S1A). Kanamycin consistently yielded more transformants than zeocin. In *M. marinum*, glycine similarly enhanced transformation by 2- and 8-fold for pQL037 and pQL039 (hygromycin resistance) (Fig. S1D). All constructs expressed mScarlet, allowing direct visualization of fluorescent colonies and eliminating false positives. Preparation of competent cells at room temperature provided a slight but consistent advantage over ice conditions and was therefore adopted in subsequent protocols (Fig. S1A and D). By contrast, the addition of DMSO, LiAc/DTT, or a 1-hour 42°C heat shock each conferred only modest improvements. None produced desirable enhancements beyond glycine pretreatment alone, nor did they exhibit synergistic effects when combined (Fig. S1B and C, S1E and F).

### Ultra-high electric field strength significantly enhances electrotransformation efficiency in nontuberculous mycobacteria

Electrotransformation efficiency is known to increase with higher electric field strength; however, this comes at the cost of increased cell death due to membrane damage. Thus, transformation efficiency reflects a tradeoff between enhanced DNA uptake and reduced viability. Mycobacteria possess thick, lipid-rich cell walls that are relatively resistant to lysis, necessitating the use of relatively high field strengths (e.g., 1.25 kV/mm, corresponding to 2.5 kV across a 2 mm cuvette) in standard protocols.

In our initial optimization experiments, we employed an electric field strength of 1.25 kV/mm. We hypothesized that further increasing the electric field strength could improve transformation efficiency in NTM. To mitigate the cytotoxic effects of ultra-high field strength, we supplemented both the growth medium and electroporation buffer with osmoprotectants: 0.5 M sorbitol in the culture medium and a combination of 0.5 M sorbitol, 0.5 M mannitol, and 10% glycerol in the electroporation solution. These hyperosmotic conditions are known to increase resistance to electrical stress (23). In *M. abscessus* ATCC 19977 strain, transformation efficiency increased progressively with electric field strength (Fig. 1A and B). At 3.0 kV/mm—the maximum output of the Bio-Rad MicroPulser—transformation efficiencies for plasmids pQL037 and pQL038 increased by 84-fold and 13-fold, respectively, relative to 1.25 kV/mm. Notably, even without osmoprotective pretreatment, the ATCC 19977 strain exhibited high tolerance to electrical damage, achieving transformation efficiencies of $3.8 \times 10^6$ CFU/µg DNA for pQL037 and $7 \times 10^4$ CFU/µg DNA for pQL038—representing 106- and 18-fold increases, respectively, over conventional conditions. In contrast, hyperosmotic pretreatment was essential for efficient transformation of a *M. abscessus* rough morphotype strain lacking glycopeptidolipid (GPL) synthesis. At 3.0 kV/mm, the GPL-deficient strain exhibited a 20-fold and 55-fold increase in transformation efficiency for pQL037 and pQL038, respectively, reaching $3 \times 10^6$ and $6 \times 10^4$ CFU/µg DNA. Without hyperosmotic treatment, gains were significantly lower (5-fold and 9-fold, respectively) (Fig. 1E).

In *M. marinum*, electrotransformation efficiency also increased with field strength. At 3.0 kV/mm, transformation with pQL037 and pQL039 reached $2.9 \times 10^8$ and $1.4 \times 10^8$ CFU/µg DNA, respectively—representing 83-fold and 29-fold increases over baseline conditions (Fig. 1C and D). In this strain, osmoprotective pretreatment had minimal

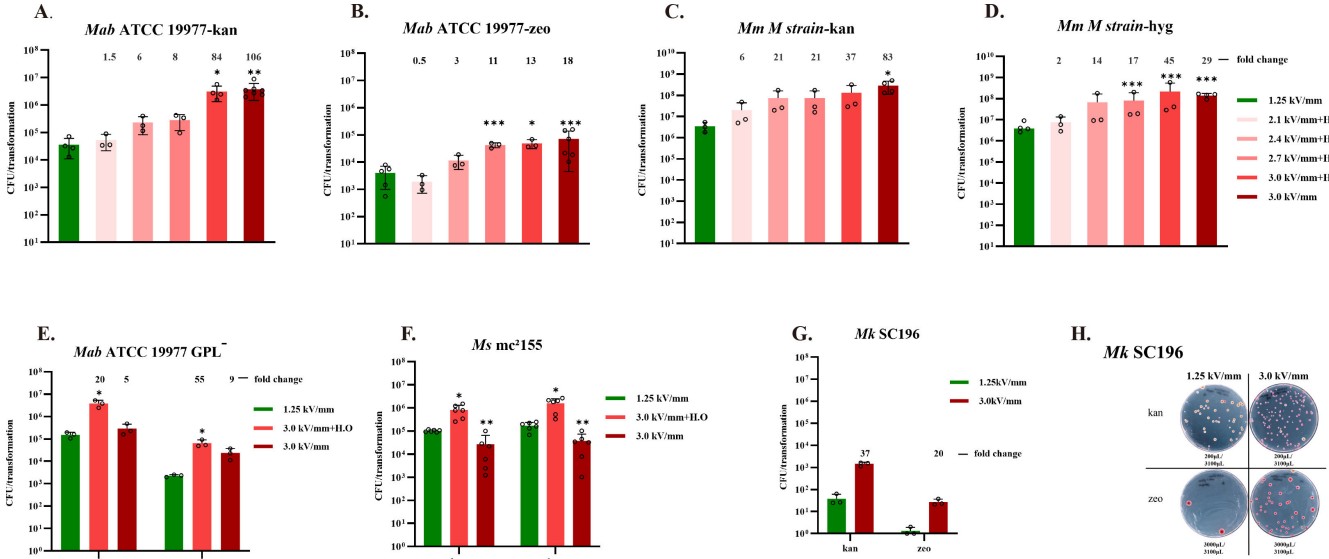

**FIG 1** Ultra-high electric field strength electroporation markedly enhances plasmid transformation efficiency in *M. abscessus*, *M. marinum*, and *M. kansasii*. (A and B) Transformation efficiency of kanamycin-resistant plasmid pQL037 (A) and zeocin-resistant plasmid pQL038 (B) in *M. abscessus* across a range of electric field strengths. Each condition used 100 µL of competent cells and 1 µg of desalted plasmid DNA. Fold increases in colony numbers relative to the conventional field strength (1.25 kV/ mm) are indicated above bars. H.O. denotes hypertonic conditions, achieved by supplementing cultures with 0.5 M sorbitol and adding 0.5 M sorbitol and 0.5 M mannitol to the electroporation buffer in addition to 10% glycerol. (C and D) Transformation efficiency of pQL037 (C) and hygromycin-resistant plasmid pQL039 (D) in *M. marinum* at varying electric field strengths. (E) Effect of ultra-high electric field strength and hypertonic protectants on transformation of pQL037 and pQL038 in a glycopeptidolipid (GPL)-deficient *M. abscessus* strain lacking the *mps2* (*mab_4098*c）gene. (F) Electroporation efficiency of pQL037 and pQL039 in *M. smegmatis mc²155* under ultra-high electric field conditions with or without hyperosmotic treatment. (G) Transformation efficiency of pQL037 and pQL038 in clinical isolate SC196 of *M. kansasii* using ultra-high field electroporation. (H) Representative image of colonies on selective medium following electroporation of *M. kansasii* SC196. Both plasmids express the fluorescent protein mScarlet; purple-red colonies indicate successful transformation. All other experiments were independently repeated three to six times. *P*-value: *<0.05; **<0.01; ***<0.001. If the data meet assumptions of homogeneity of variance and normality, use the independent-samples *t*-test. If variances are unequal, use the Welch *t*-test. If normality is violated, use the Mann–Whitney *U* test.

impact at ultra-high field strengths. In contrast, *M. smegmatis* mc²155 was highly sensitive to electric field–induced damage. Transformation efficiency at 3.0 kV/mm dropped markedly and exhibited high interexperimental variability (Fig. 1F). However, hyperosmotic pretreatment markedly improved tolerance: transformation efficiencies for pQL037 and pQL039 increased by 8-fold and 9-fold, respectively, over those achieved at 1.25 kV/mm. The electrotransformation efficiency of *M. kansasii* SC196 for exogenous plasmid DNA is notably low under standard conditions, yielding only 64 CFU/µg DNA for plasmid pQL037 and 2 CFU/µg DNA for pQL038. This inefficiency may be attributable to the presence of the *eptC* gene, which encodes a protein that impedes plasmid segregation during cell division (24). Application of an ultra-high electric field strength (3.0 kV/mm) also significantly enhanced transformation efficiency, resulting in $1.5 \times 10^3$ CFU/µg DNA for pQL037 and 36 CFU/µg DNA for pQL038 (Fig. 1G and H). We also assessed whether combining ultra-high electric field strength (3 kV/mm) with DMSO, LiAc/DTT, or heat shock (42°C) could further enhance transformation efficiency. However, these combinatorial treatments conferred no additional benefit beyond that achieved with high-voltage electroporation alone (Fig. S2A and B)

Taken together, these results demonstrate that ultra-high electric field strength (3.0 kV/mm) substantially enhances electrotransformation efficiency across multiple NTM species, including *M. abscessus*, *M. marinum*, and *M. kansasii*. The requirement for osmoprotective pretreatment appears to be strain-specific, with cell wall–deficient strains such as GPL-deficient *M. abscessus* and *M. smegmatis* mc²155 benefiting most from hyperosmotic conditions.

## Ultra-high electric field electroporation significantly enhances gene editing efficiency in *M. abscessus*

The RecET recombination system derived from mycobacteriophage Che9c has been shown to mediate efficient gene knockout, reporter fusion, and point mutation editing in *M. tuberculosis* and *M. smegmatis* (12, 13). However, its recombination efficiency in *M. abscessus* remains extremely low, with few antibiotic-resistant colonies recovered and only approximately 7% of those exhibiting correct allele exchange (14). This inefficiency is likely attributable, in part, to the inherently low electrotransformation efficiency of *M. abscessus*.

To assess whether increased transformation efficiency could enhance recombination, we leveraged ultra-high electric field strength electroporation (3.0 kV/mm), previously shown to boost plasmid delivery efficiency by >100-fold in *M. abscessus*. Six genes (*mmpl4*, *mmpl6*, *mmpl7*, *prcA*, *uvrB*, and *recD*) were selected as targets. The RecET-expressing vector pJV53 was modified to carry a zeocin resistance gene and an mScarlet fluorescent marker (pJV53-zeo-mScarlet) and then electroporated into wild-type *M. abscessus* ATCC 19977. Allelic exchange substrates, consisting of approximately 1 kb homology arms flanking a *kanR* cassette, were introduced under conventional or ultra-high field electroporation. As shown in Fig. 2A through F, ultra-high field electroporation yielded dramatically increased numbers of kanR colonies—1580-, 1098-, 680-, 1102-, 653-, and 721-fold greater than conventional electroporation for the six respective targets, all exceeding $10^5$ colonies per transformation. In contrast, conventional electroporation yielded approximately 100 colonies per transformation. PCR verification of 23 randomly selected kanR colonies revealed correct allele exchange in only 13–43% of conventional transformants, whereas nearly all colonies from the ultra-high field group exhibited correct editing (22/23 for *mmp16*; 23/23 for the others; Fig. 2A through F, Fig. S3A and B). These results indicate that improved uptake of linear DNA substrates via ultra-high field electroporation substantially enhances homologous recombination efficiency in *M. abscessus*.

To evaluate single-base gene editing, we adapted an oligonucleotide-mediated recombineering system using the Che9c RecT protein. A 70-nt oligo designed to repair an inactivating E118* mutation in the *kanR* gene was electroporated into *M. abscessus* expressing RecT from the pKM461-dkan-ZeoR plasmid. As expected, the lagging-strand-targeting Oligo1 yielded approximately 7-fold more kanR colonies than the leading-strand Oligo2 (Fig. 2G and H). Notably, ultra-high field electroporation improved the yield of kanR colonies by >1,500-fold for both oligos compared to conventional electroporation. We further applied this system to introduce a clinically relevant single-nucleotide mutation (c.288C > T; Asp96Asn) into *gyrA*, conferring fluoroquinolone resistance (25). Electroporation of Oligo-gyrAAsp96Asn into RecT-expressing *M. abscessus*, followed by selection on ofloxacin (80 µg/mL) plates, produced ~$5 \times 10^7$ CFU/2 µg oligo with ultra-high field electroporation, >10,000-fold higher than conventional electroporation (Fig. 2I). Sanger sequencing of eight colonies from each group revealed correct point mutation incorporation in all ultra-high field clones and only two of eight from the conventional group (Fig. 2J).

Together, these results demonstrate that ultra-high electric field strength electroporation substantially improves both double-stranded DNA and oligonucleotide-mediated gene editing efficiency in *M. abscessus*.

## DISCUSSION

Efficient electrotransformation and gene editing in NTM, such as *M. abscessus*, remain longstanding technical challenges. Existing protocols for electrotransformation in *M. abscessus* are largely extrapolated from work in *M. smegmatis* and *M. tuberculosis*, where key factors—including growth phase, glycine pretreatment, low-temperature cell preparation, and desalting of DNA and cells—have been shown to influence transformation efficiency. Recent efforts by Campo-Pérez et al. introduced a 42°C heat shock step to

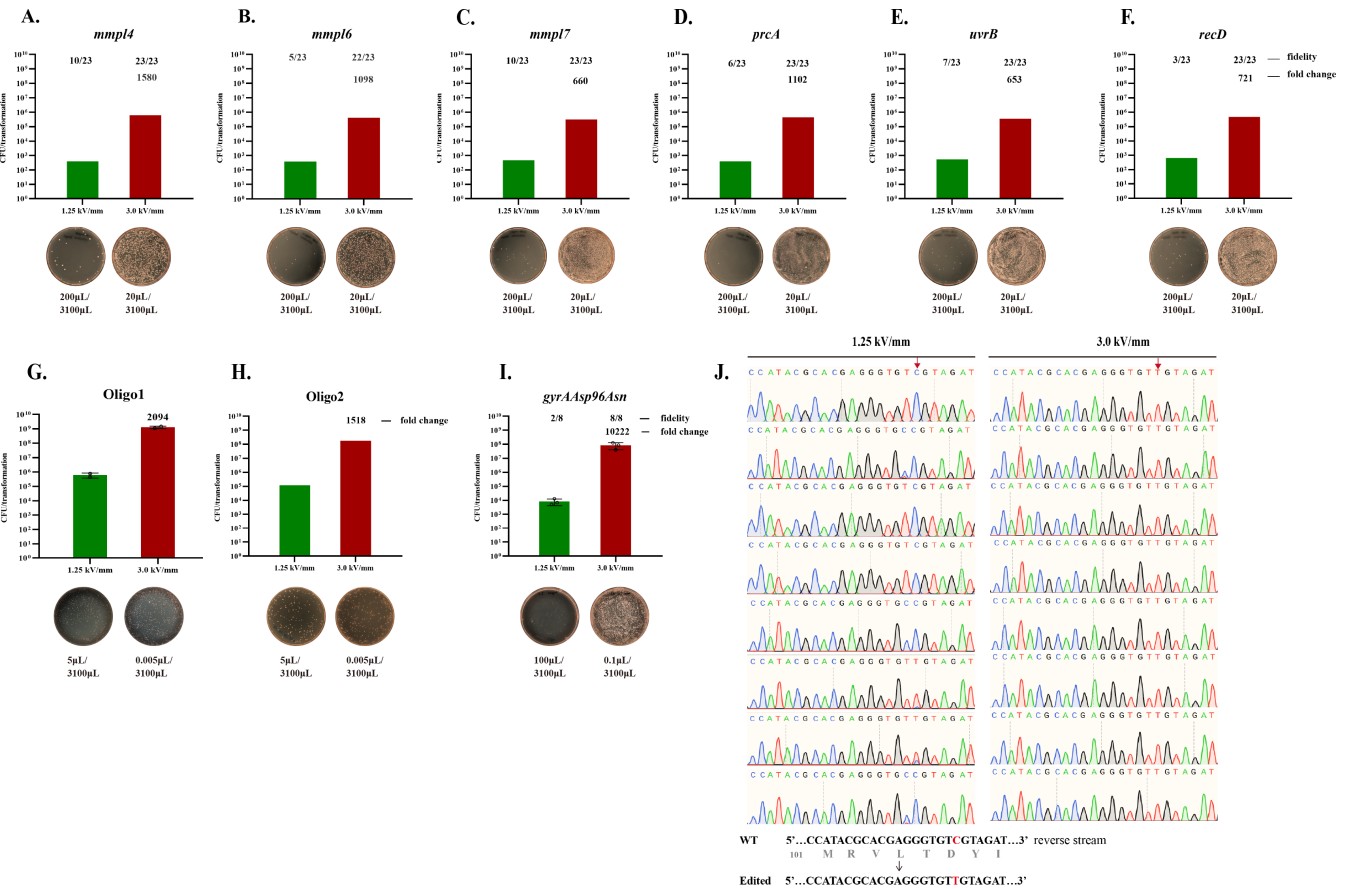

**FIG 2** Ultra-high electric field strength electroporation enhances recombination efficiency in *M. abscessus*. (A through F) Gene knockout efficiency is significantly increased using ultra-high electric field strength electroporation of allele exchange substrates in the presence of RecET recombinase. Wild-type *M. abscessus* carrying pJV53-zeo-mScarlet was grown to log phase and induced with 0.2% acetamide to express RecET prior to competent cell preparation. A total of 200 ng of allele exchange substrates targeting six loci—*mmpl4* (A), *mmpl6* (B), *mmpl7* (C), *prcA* (D), *uvrB* (E), and *recD* (F)—were introduced by electroporation under either ultra-high or conventional electric field conditions. Transformants were selected on kanamycin-containing medium. Fold changes in colony numbers (ultra-high vs conventional field strength) are indicated above each bar. For each gene, 23 kanamycin-resistant colonies were randomly selected for verification by colony PCR; successful knockout rates are shown above each bar. Representative selection plates are shown below, with plated culture volumes indicated. (G and H) Oligonucleotide recombineering was assessed using 2 µg of 70-nt oligos (Oligo1, lagging strand; Oligo2, leading strand) targeting a kanamycin repair site in *M. abscessus* harboring the pKM461-dkan-ZeoR plasmid. Electroporation was performed under ultra-high or conventional field strengths, and kanamycin-resistant transformants were enumerated. Fold increases in colony numbers and representative selection plates are shown; plated culture volumes are indicated. (I) Point mutation recombineering was performed using 2 µg of a 70-nt oligonucleotide (Oligo-gyrAAsp96Asn) introducing a c.288C > T substitution in the *gyrA* gene. Electroporation into *M. abscessus* expressing RecT was followed by selection on ofloxacin (80 µg /mL). Fold enrichment and representative plates are shown. (J) Eight colonies from each condition in (I) were analyzed by Sanger sequencing to determine editing accuracy at the *gyrA* locus.

suppress restriction activity and disperse cell aggregates, modestly improving *M. abscessus* electrotransformation efficiency ($<5 \times 10^4$ CFU/µg DNA) (10). Consistent with earlier findings, we observed that glycine pretreatment substantially enhanced plasmid delivery efficiency in both *M. abscessus* and *M. marinum*. Further enhancements were achieved with 42°C heat shock or chemical treatments that increase membrane permeability (e.g., DMSO, LiAc/DTT), though the improvements remained limited. In contrast, application of ultra-high electric field strength (3.0 kV/mm) during electroporation led to a striking increase in transformation efficiency—up to $2.4 \times 10^6$ CFU/µg DNA for *M. abscessus* and $3.3 \times 10^8$ CFU/µg DNA for *M. marinum*.

The success of electroporation hinges on the balance between enhancing membrane permeability and preserving cell viability. While increasing electric field strength promotes DNA uptake, it concurrently exacerbates membrane damage. In *Escherichia coli*, this trade-off manifests as a bell-shaped relationship, with transformation efficiency

peaking at approximately 1.2 kV/mm and declining thereafter due to compromised survival (8). By contrast, we observed a monotonic increase in electroporation efficiency in *M. abscessus* and *M. marinum* across a tested range of 1.25 to 3.0 kV/mm (3.0 kV is the maximum voltage capacity of the Bio-Rad Pulsers), suggesting a higher threshold for electroporation-induced lethality in these species. This apparent resilience may reflect fundamental differences in cell envelope composition between these mycobacteria and other bacteria. Notably, NTM such as *M. abscessus* possess a thick, lipid-rich outer membrane that likely serves as a barrier to electropermeabilization-induced rupture. Our data further support this hypothesis: strains deficient in GPLs—including *M. abscessus* GPL mutants and *M. smegmatis* mc$^2$155—were markedly more susceptible to electroporation at high voltages. The absence of key surface lipids, such as lipooligosaccharides and polar GPLs (as in mc$^2$155) (26), likely disrupts the envelope's mechanical integrity, rendering cells more vulnerable to electric field-induced stress.

These findings align with previous reports in Gram-positive species, where the inclusion of osmoprotectants, such as sorbitol or mannitol, enhances electroporation survival by mitigating osmotic shock (23, 27, 28). In our hands, the addition of hypertonic agents substantially improved transformation efficiencies in GPL-deficient *M. abscessus* and *M. smegmatis* mc$^2$155 strains. However, for wild-type *M. abscessus* and *M. marinum*, these additives conferred little additional benefit, suggesting that their native envelope structure is sufficient to withstand high-voltage pulses.

Arcing is a well-recognized complication during electroporation at elevated voltages, often exacerbated by residual salts. Despite applying electric field strengths up to 3.0 kV/mm, we observed no notable arcing events. This may reflect stringent salt control—achieved by dissolving plasmid DNA in deionized water and thoroughly washing competent cells—as well as our use of a reduced electroporation volume (100 µL in 0.1 cm gap cuvettes), which minimizes conductive load compared to standard protocols (400 µL in a 0.2 cm cuvette). It also should be noted that the choice of electroporation device also impacted pulse dynamics. With the Bio-Rad MicroPulser, time constants ranged from 5 to 7 ms across voltages tested (1.25–3.0 kV), providing efficient delivery with minimal arcing. In contrast, the Bio-Rad GenePulser (configured with a 1,000-ohm resistance) produced longer pulse durations (15–25 ms) and higher arcing frequency in our hands.

In our experiments, *M. abscessus* yielded markedly fewer transformants when electroporated with plasmids encoding zeocin resistance compared to otherwise identical plasmids carrying kanamycin resistance. This consistent disparity is likely attributable to fundamental differences in the pharmacodynamics and resistance mechanisms of the two antibiotics. Zeocin, a member of the bleomycin family, exerts bactericidal activity by inducing rapid DNA double-strand breaks, leading to cell death before sufficient resistance gene expression can occur. In contrast, kanamycin disrupts protein synthesis more gradually, affording a longer window for the *aph* gene to be transcribed and translated post-electroporation. Thus, kanamycin's delayed cytotoxicity may facilitate higher recovery rates of transformants. The biochemical nature of the resistance mechanisms may further amplify this difference. The *aph* gene encodes aminoglycoside phosphotransferase (APH), an enzyme that catalytically inactivates kanamycin via phosphorylation. A single APH molecule can detoxify multiple antibiotic molecules, providing a robust and amplifiable defense. By contrast, zeocin resistance relies on the *Streptoalloteichus hindustanus* (*Sh*) *ble* gene, which encodes a 14 kDa protein that binds zeocin in a stoichiometric manner—each Sh Ble molecule sequesters only a single zeocin molecule. As a result, effective resistance requires high intracellular concentrations of Sh Ble, which may not be rapidly achieved following transformation. This distinction between catalytic and stoichiometric detoxification mechanisms likely underpins the reduced efficiency observed with zeocin-resistant constructs. Moreover, prior studies have suggested that even in the presence of stably expressed Sh Ble protein, zeocin may retain residual genotoxic activity, continuing to damage DNA in

mammalian systems (29). A similar effect in mycobacteria could further compromise cell viability during the critical post-electroporation recovery period.

The low efficiency of gene editing in *M. abscessus* has been a major bottleneck in advancing molecular genetics in this pathogen. Although CRISPR/Cas9 systems have recently enabled efficient gene inactivation (30, 31), precise genome editing—such as gene deletions, point mutations, and reporter insertions—still depends heavily on homologous recombination systems. Yet, the Che9c-derived RecET system, highly efficient in *M. smegmatis* and *M. tuberculosis*, performs poorly in *M. abscessus*. This is especially true for oligonucleotide-mediated recombination, where efficiency is below 1 in 1,000,000 (16), rendering it nearly impractical for generating point mutation strains. In this study, we demonstrate that delivering recombination substrates via ultra-high electric field strength electroporation overcomes this limitation. Gene knockout efficiency was enhanced by nearly 1,000-fold, and oligonucleotide-mediated point mutation efficiency improved by >10,000-fold. These findings strongly suggest that the previously observed poor recombination in *M. abscessus* is due, at least in part, to inefficient uptake of DNA substrates. While our current study focused on *M. abscessus*, we observed similar improvements in plasmid uptake in *M. marinum*, implying that ultra-high field electroporation may also enhance homologous recombination in other NTM species. Future work should evaluate whether this strategy can broadly improve gene editing efficiency across diverse mycobacterial species, including slow growers and clinical isolates.

In summary, we show that *M. abscessus* and other NTM species possess an unusual tolerance to ultra-high electric field strengths, likely conferred by their robust cell envelope. Exploiting this trait, we developed a highly efficient electroporation protocol that dramatically improves transformation and gene editing efficiency. In addition to enabling precise genome engineering, this method may facilitate high-throughput applications, such as CRISPRi screening libraries and plasmid expression systems in mycobacteria. Further exploration of this approach in other mycobacterial models may expand its utility and uncover new insights into bacterial electrobiology.

## ACKNOWLEDGMENTS

This work was supported by grants from the Noncommunicable Chronic Diseases-National Science and Technology Major Project (No. 2024ZD0528400) and the National Natural Science Foundation of China (Grant No. 82272375).

The authors declare that the research was conducted in the absence of any commercial or financial relationships that could be construed as a potential conflict of interest.

## AUTHOR AFFILIATIONS

[1]Department of Respiratory and Critical Care Medicine, West China Hospital, Sichuan University, Chengdu, China
[2]Institute of Respiratory Health, Frontiers Science Center for Disease-related Molecular Network, West China Hospital, Sichuan University, Chengdu, China
[3]Guangdong Provincial Key Laboratory of Infection Immunity and Inflammation, Department of Pathogen Biology, Shenzhen University Medical School, Shenzhen, China
[4]Department of Pathogen Biology, West China School of Basic Medical Sciences & Forensic Medicine, Sichuan University, Chengdu, China
[5]State Key Laboratory of Respiratory Health and Multimorbidity, West China Hospital, Sichuan University, Chengdu, China

## AUTHOR ORCIDs

Daoyan Tang http://orcid.org/0009-0007-6215-4055
Yi Cai http://orcid.org/0000-0002-1363-2328
Tao Luo http://orcid.org/0000-0002-2929-2233
Jianqing He http://orcid.org/0000-0002-4214-2037

Qinglan Wang http://orcid.org/0000-0002-7629-5058

## FUNDING

| Funder | Grant(s) | Author(s) |
|---|---|---|
| Noncommunicable Chronic Diseases-National Science and Technology Major Project of China | No. 2024ZD0528400 | Qinglan Wang |
| National Natural Science Foundation of China | No. 82272375 | Qinglan Wang |

## AUTHOR CONTRIBUTIONS

Daoyan Tang, Data curation, Formal analysis, Writing – original draft | Minggui Wang, Data curation, Formal analysis, Writing – original draft | Dan Wang, Data curation | Danni Yang, Data curation | Yi Cai, Formal analysis, Methodology | Tao Luo, Formal analysis, Methodology | Jianqing He, Conceptualization, Supervision | Qinglan Wang, Conceptualization, Formal analysis, Funding acquisition, Methodology, Project administration, Supervision, Writing – original draft

## DATA AVAILABILITY

All data supporting the findings of this study are available within the paper and its supplemental material file.

## ADDITIONAL FILES

The following material is available online.

### Supplemental Material

**Supplemental figures and tables (Spectrum01944-25-S0001.pdf).** Fig. S1 and S2, and Tables S1 and S2.

### Open Peer Review

**PEER REVIEW HISTORY (review-history.pdf).** An accounting of the reviewer comments and feedback.

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
