## [Reviewer comments · Microbiology Spectrum]

Microbiology Spectrum

Ultra-high field strength electroporation enables efficient DNA transformation and genome editing in nontuberculous mycobacteria

Daoyan Tang, Ming-Gui Wang, Dan Wang, Danni Yang, Yi Cai, Tao Luo, Jian-qing He, and Qinglan Wang

Corresponding Author(s): Qinglan Wang, Sichuan University

Review Timeline:

Submission Date:	July 2, 2025
Editorial Decision:	July 25, 2025
Revision Received:	July 28, 2025
Accepted:	August 5, 2025

Editor: Silvia Cardona

Reviewer(s): The reviewers have opted to remain anonymous.

Transaction Report:

DOI: <https://doi.org/10.1128/spectrum.01944-25>

Re: Spectrum01944-25 (**Ultra-high field strength electroporation enables efficient DNA transformation and genome editing in nontuberculous mycobacteria**)

Dear Prof. Qinglan Wang:

I am pleased to inform you that your manuscript will be accepted upon minor corrections. I am sharing the latest comments by reviewers. Please consider incorporating the suggested changes. Upon re-submission, your manuscript will be formally accepted.

Please return the manuscript within 30 days; if you cannot complete the modification within this time period, please contact me. If you do not wish to modify the manuscript and prefer to submit it to another journal, notify me immediately so that the manuscript may be formally withdrawn from consideration by Spectrum.

Revision Guidelines

Sincerely,
Silvia Cardona
Editor
Microbiology Spectrum

Reviewer #1 (Comments for the Author):

The authors report several technical tweaks, and especially the increase in voltage, to increase the transformation efficacy in several mycobacteria, and especially *M. abscessus*.

Increasing the efficacy in *M. abscessus* is an important finding, and is definitely worth publication.

I have reviewed the previous version of the paper. At the time I pointed to many unclear points and parts. These, I am happy to

say, were mostly corrected.

I additionally suggested that most of the technical tweaks (other than the change in electroporation voltage which appears to be the key here) be mostly removed from the manuscript. The authors have made these parts a bit smaller, but still present them. I think it is their right to do so, but again - my personal advice would be to minimize these even further, as their effect proved to be small, and none of them was novel per-se. I believe that when a true technical advance is presented, it is better to present it "to the point", without additional data that make the reader wonder what parts are the essence, and which parts are only making the work complicated, but with little, if any benefit.

However, I do leave this to the editor discretion to ask (or not to ask) the authors to follow my line of thinking. Otherwise, the manuscript is nice, and in principle - worthy of publication.

Reviewer #2 (Comments for the Author):

All my concerns have been addressed. The MS can be accepted now.

Reviewer #3 (Comments for the Author):

The study by Tang et al. reports a methodological and systematic approach to building upon previously established electroporation protocols and in essence, improving upon them by increasing the electric field strength. By establishing known electroporation parameters from previous studies, the authors were able to show reproducibility and demonstrate significant improvements in transformant yield by increasing the electric field strength from 1.25 kV to 3.0 kV/mm. In this manuscript version, the authors successfully responded to each of the reviewers' comments by revising the text where appropriate which significantly improved the strength and impact of this technique-driven manuscript to the non-tuberculosis mycobacteria research field.

A couple of minor comments for the authors to consider:

1. Figure 1 legend: please include explanations for RT and ice for reader's benefit. Likewise, for R-DMSO.

Dear Editor and Reviewers,

We sincerely thank the Editor and Reviewers for their thoughtful and constructive comments on our manuscript entitled “*Ultra-high field strength electroporation enables efficient DNA transformation and genome editing in nontuberculous mycobacteria*” (Article ID: Spectrum01944-25).

We have carefully revised the manuscript in response to the suggestions provided and believe the changes have significantly improved the clarity and overall quality of the work.

Please find below our detailed, point-by-point responses to each comment. We hope that our revisions address all concerns satisfactorily, and we are grateful for the opportunity to resubmit our work for further consideration.

Sincerely yours,

Qinglan Wang

West China Hospital, Sichuan University,

E-mail: wangqinglan@scu.edu.cn.

Response to Reviewer Comments and Editorial Requests

We thank the reviewers for their overall enthusiasm and helpful comments, which have helped improve the manuscript. Author responses are provided below in blue font. Revised sections of the manuscript (Marked Up Manuscript) are also highlighted in blue.

Reviewer 1:

The authors report several technical tweaks, and especially the increase in voltage, to increase the transformation efficacy in several mycobacteria, and especially *M. abscessus*.

Increasing the efficacy in *M. abscessus* is an important finding, and is definitely worth publication.

I have reviewed the previous version of the paper. At the time I pointed to many unclear points and parts. These, I am happy to say, were mostly corrected.

I additionally suggested that most of the technical tweaks (other than the change in electroporation voltage which appears to be the key here) be mostly removed from the manuscript. The authors have made these parts a bit smaller, but still present them. I think it is their right to do so, but again - my personal advice would be to minimize these even further, as their effect proved to be small, and none of them was novel per-se. I believe that when a true technical advance is presented, it is better to present it "to the point", without additional data that make the reader wonder what parts are the essence, and which parts are only making the work complicated, but with little, if any benefit. However, I do leave this to the editor discretion to ask (or not to ask) the authors to follow my line of thinking.

Otherwise, the manuscript is nice, and in principle - worthy of publication.

R1 We sincerely thank the reviewer for the thoughtful and encouraging comments on our work. We fully agree with the reviewer's assessment that the increase in electroporation voltage constitutes the primary technical advance, with the most pronounced and reproducible impact on transformation efficiency in NTM, as reflected in the title and central message of the manuscript.

In line with the reviewer's suggestion to streamline the presentation of less impactful optimization conditions, we have moved the entirety of Figure 1—covering cell wall-weakening pretreatments—to the Supplementary Material and have substantially reduced the associated text in the main manuscript. The revised paragraph now briefly states as shown in Section 1 of Results

Reviewer 2:

All my concerns have been addressed. The MS can be accepted now.

We sincerely thank the reviewer for the positive feedback and are pleased to hear that all concerns have been fully addressed. We greatly appreciate the reviewer's time and thoughtful input, which helped improve the quality and clarity of our manuscript.

Reviewer 3:

The study by Tang et al. reports a methodological and systematic approach to building upon previously established electroporation protocols and in essence, improving upon them by increasing

the electric field strength. By establishing known electroporation parameters from previous studies, the authors were able to show reproducibility and demonstrate significant improvements in transformant yield by increasing the electric field strength from 1.25 kV to 3.0 kV/mm. In this manuscript version, the authors successfully responded to each of the reviewers' comments by revising the text where appropriate which significantly improved the strength and impact of this technique-driven manuscript to the non-tuberculosis mycobacteria research field.

We thank the reviewer for the positive and encouraging feedback on our work. We are especially grateful for the recognition of the reproducibility and impact of our optimized electroporation protocol for nontuberculous mycobacteria, and we appreciate the reviewer's acknowledgement of the improvements made in the revised manuscript.

A couple of minor comments for the authors to consider:

Q1 Figure 1 legend: please include explanations for RT and ice for reader's benefit. Likewise, for R-DMSO.

R1 We thank the reviewer for pointing this out. As suggested by Reviewer 1, the original Figure 1 has been moved to the Supplementary Material as Figure S1. In the revised legend for Figure S1, we have now clarified all abbreviations as follows:

R.T.: refers to room temperature (~25 °C);

Ice: refers to cultures placed on ice for 1 h prior to centrifugation, with all subsequent steps also performed at 4 °C (centrifugation) or on ice (cell handling) during competent cell preparation;

2% R-DMSO: refers to recovery medium supplemented with 2% (v/v) dimethyl sulfoxide (DMSO).

We hope these clarifications enhance the clarity and accessibility of the figure for readers.

Re: Spectrum01944-25R1 (**Ultra-high field strength electroporation enables efficient DNA transformation and genome editing in nontuberculous mycobacteria**)

Dear Prof. Qinglan Wang:

Your manuscript has been accepted, and I am forwarding it to the ASM production staff for publication. Your paper will first be checked to make sure all elements meet the technical requirements. ASM staff will contact you if anything needs to be revised before copyediting and production can begin. Otherwise, you will be notified when your proofs are ready to be viewed.

Sincerely,
Silvia Cardona
Editor
Microbiology Spectrum